# Modeling and Experimental Verification of Time-Controlled Grinding Removal Function for Optical Components

**DOI:** 10.3390/mi14071384

**Published:** 2023-07-06

**Authors:** Fulei Chen, Xiaoqiang Peng, Zizhou Sun, Hao Hu, Yifan Dai, Tao Lai

**Affiliations:** 1College of Intelligent Science and Technology, National University of Defense Technology, Changsha 410073, China; ch_fl5711@163.com (F.C.);; 2Hunan Key Laboratory of Ultra-Precision Machining Technology, Changsha 410073, China; 3Laboratory of Science and Technology on Integrated Logistics Support, National University of Defense Technology, Changsha 410073, China

**Keywords:** time-controlled grinding, abrasive belt, removal function, optical components, deterministic machining

## Abstract

As a flexible grinding method with high efficiency, abrasive belt grinding has been widely used in the machining of mechanical parts. However, abrasive belt grinding has not been well applied in the field of ultra-precision optical processing, due to the lack of a stable and controllable removal function. In this paper, based on the idea of deterministic machining, the time-controlled grinding (TCG) method based on the abrasive belt as a machining tool was applied to the deterministic machining of optical components. Firstly, based on the Preston equation, a theoretical model of the TCG removal function was established. Secondly, removal function experiments were carried out to verify the validity and robustness of the theoretical removal model. Further, theoretical and actual shaping experiments were carried out on 200 mm × 200 mm flat glass-ceramic. The results show that the surface shape error converged from 6.497 μm PV and 1.318 μm RMS to 5.397 μm PV and 1.115 μm RMS. The theoretical and experimental results are consistent. In addition, the surface roughness improved from 271 to 143 nm Ra. The results validate the concept that the removal function model established in this paper can guide the actual shaping experiments of TCG, which is expected to be applied to the deterministic machining of large-diameter optical components.

## 1. Introduction

Large-aperture optical systems are widely used in high-tech fields such as space science research, astronomical exploration, and Earth observations [1,2,3]. Currently, optical systems are moving toward larger apertures and higher precision, which places further demands on the processing of optical components [4]. The general machining process for large-diameter optical components is grinding, lapping, and polishing. After the first two processes converge the surface shape error of several tens of microns to better than 1–2 μm, the final processing stage is achieved by connecting the polishing process. The grinding stage accounts for more than 60% of the total time spent in the process. Improving the accuracy of the grinding stage, and thus eliminating or reducing the time of the lapping stage, is considered to be one of the most efficient and economical methods to machine large-diameter optical components [5].

In the field of optical manufacturing, conventional grinding mostly uses grinding wheels as grinding tools, which have the advantage of high material removal efficiency. However, wheel grinding is overly dependent on machine tool accuracy, and problems such as grinding wheel wear, grinding wheel clogging, and grinding burns [6,7] also limit further improvements in its processing accuracy. In addition, to obtain excellent properties such as high strength, a light weight, corrosion resistance, and a low coefficient of thermal expansion, the materials of large-diameter optical components are mostly hard and brittle materials such as glass-ceramic and silicon carbide. These materials have the characteristics of low fracture toughness and poor ductility and are typically difficult to machine [8]. In order to achieve the high-efficiency and high-quality grinding of glass-ceramic, ultrasonic vibration-assisted grinding (UVAG) has been of general interest to researchers in recent years. A series of studies have been conducted to investigate the effects of grinding force, grinding method, and grinding temperature on the material removal efficiency and surface roughness of UVAG [9,10,11]. However, the prediction of the material removal rate in UVAG is still a complex problem [12]. Therefore, it is difficult to achieve the micron-level surface accuracy of large-diameter optical components with conventional grinding wheels [13].

Abrasive belt grinding is a flexible grinding method with high efficiency, low energy consumption, and low cost [14]. Compared with wheel grinding, abrasive belt grinding requires less machine tool stiffness. Therefore, abrasive belt grinding has great potential for the machining of large-diameter optical components. OptiPro Systems of the United States and the University of Rochester proposed a kind of wheel belt deterministic polishing technology, named UniForm Finishing (UFF) [15]. Fess et al. [16] used UFF to process a sapphire window in 70 min, and the surface shape error converged from RMS 0.18 λ to RMS 0.028 λ. Liu et al. [17] proposed a two-dimensional elliptical vibrating abrasive belt grinding (2D-EVBG) technology. The machining results show that lower surface roughness will be obtained with the increase in grinding depth and workpiece feed speed, which is also more likely to lead to brittleness removal for quartz glass. Ren et al. [18] studied the prediction method of material removal depth contour (MRDC) based on material removal rate. The unit material removal volume was calculated as the cross-section area multiplied by the grinding length per unit of time. The model achieved good prediction accuracy on the GCr15 material. Wang et al. [19] proposed a method for the online monitoring of abrasive belt material removal rates and their corresponding wear statuses using grinding sound signals. The model achieved better results on metallic materials. Yan et al. [14] developed an improved cutting force model to analyze and assess force-controlled robotic abrasive belt grinding mechanisms. The practicality and validity of the force model was verified by taking a robot belt grinding test workpiece, a Ti-6Al-4V alloy, as an example. However, there are few studies on abrasive belt grinding in the field of optical processing, mostly in the field of mechanical processing, and the studies are mostly focused on the grinding force and grinding depth. In addition, the current abrasive belt grinding technique is mostly non-deterministic processing, which limits its development in the field of optical processing with high requirements for processing quality.

TCG is a computer-controlled optical surface (CCOS)-forming technology based on the deterministic shaping method. Removal function modeling can effectively predict a material’s removal characteristics during its grinding process, which facilitates the regulation of the removal function for the better guidance of machining. It is of great significance for the realization of TCG with high determinism. 

Therefore, this study applies TCG with abrasive belts as machining tools to the deterministic machining of optical components based on the idea of CCOS deterministic machining. The research idea of this paper is shown in Figure 1. Firstly, the theoretical model of the removal function of TCG is established based on the Preston equation via MATLAB 2017a. Secondly, the removal function experiments are carried out to verify the validity of the theoretical removal model. Finally, theoretical and actual shaping experiments are carried out to evaluate the experimental results and discuss the guiding significance of the theoretical model for practical machining.

## 2. Principle of TCG

TCG is a CCOS-forming technique based on the deterministic shaping method. The deterministic shaping method was proposed by Itek in the 1970s [20]. Based on the deterministic removal function and the initial surface shape error, the residence time of the removal function at each position on the workpiece surface is solved by an accurate residence time algorithm. The CNC control system controls the dwell time at different locations to achieve different removal amounts, thus achieving the deterministic convergence of the shape error of the optical components [21].

Unlike conventional belt grinding and UVAG, TCG satisfies the Preston equation by precisely controlling the machining parameters [22]:(1)H=KPVT
where *H* is the amount of material removed; *K* is a proportionality constant related to the environment, removal medium, material, etc.; *P* is the contact pressure between the removal tool and the workpiece; *V* is the relative velocity between the removal tool and the workpiece; and *T* is the residence time of the removal tool at one location. From Equation (1), when controlling the *K*, *P*, and *V* constants, only the dwell time *T* needs to be controlled to achieve different volumes of material removal.

The principle of TCG based on the abrasive belt as a processing tool is shown in Figure 2. The technology uses open belt grinding, where the abrasive belt is located between the contact wheel and the workpiece. During operation, the contact wheel drives the abrasive belt to vibrate at a certain amplitude and frequency in the direction of the contact wheel axis to achieve material removal from the workpiece surface. The real-time stabilization of the removal function is achieved by updating the abrasive belt at a certain speed during processing, controlling the vibration speed of the contact wheel and the constant pressure of the cylinder acting on the contact wheel.

## 3. Theoretical Modeling of Removal Function

### 3.1. Velocity Distribution in Material Removal Area

Based on the kinematic analysis, the velocity distribution in the material removal area of the TCG is shown in Figure 3, where point O is a point in the material removal area. The combined velocity *V* of the abrasive belt grain at any instant can be expressed as:(2)$$V=\sqrt{V_{Z}{}^{2}+{\left(V_{S}\pm V_{W}\right)}^{2}}$$
where *V_Z_* is the contact wheel vibration speed, *V_S_* is the belt update speed, and *V_W_* is the contact wheel movement speed during the actual grinding process. When the direction of the belt update speed in the material removal area is the same as the direction of the contact wheel movement, the plus sign is used in Equation (2). When the direction of the belt update speed in the material removal area is opposite to the direction of the contact wheel movement, the minus sign is used in Equation (2).

The belt update speed *V_S_* is directly controlled by the servo motor and reducer. The whole device moves with the machine axes, and the contact wheel movement speed *V_W_* is the machine feed speed. The contact wheel vibration speed *V_Z_* comes from the axial vibration of the contact wheel, which is realized by the eccentric mechanism with the linear guide. Its mechanism schematic diagram is shown in Figure 4. Point A is powered by the motor, and the combined velocity along the axial component (*V_Z_*) at point B drives the contact wheel vibration.

From Figure 4, the contact wheel vibration speed *V_Z_* can be expressed as:(3)$$V_{Z}=V_{ZH}\cdot \cos(\theta )$$
where *V_ZH_* is the combined velocity at point B, and *θ* is the rotation angle of the eccentric mechanism.
(4)$$V_{ZH}=\omega \cdot r$$
(5)ω=2πf
where *ω* is the angular velocity at point A, *r* is the eccentricity distance, and *f* is the vibration frequency of the contact wheel. According to Equation (2), combined with Equations (3)–(5), the combined velocity *V* of the abrasive belt grains at any instant can be obtained as follows:(6)$$V=\sqrt{{\left(2\pi fr\cos(\theta )\right)}^{2}+{\left(V_{S}\pm V_{W}\right)}^{2}}$$

As a result, the velocity variation in abrasive belt grains in one vibration cycle can be derived as shown in Figure 5.

During the actual machining process, along with the vibration of the contact wheel, the position of the contact area in the material removal area changes continuously with the rotation angle of the eccentric mechanism. The position of the eccentric mechanism corresponding to the contact area in the material removal area during one vibration cycle is shown in Figure 6. As shown in Figure 6a,c, when *θ* = 0° and *θ* = 180°, *V_Z_* = *V_ZH_*, the contact wheel is located in the middle of the vibration, and at this time, the corresponding contact area is also in the middle of the material removal area. As shown in Figure 6b, when *θ* = 90°, *V_Z_* = 0, the contact wheel is located at the leftmost position of the vibration, which corresponds to the contact area which is also at the leftmost position of the material removal area. As shown in Figure 6d, when *θ* = 270°, *V_Z_* = 0, the contact wheel is located at the rightmost position of the vibration, which corresponds to the contact area which is also at the rightmost position of the material removal area. The following relationship exists between the length of the material removal area, the width of the contact wheel, and the eccentricity distance:(7)$$L_{R}=L_{W}+2r$$
where *L_R_* is the length of the material removal area, and *L_W_* is the width of the contact wheel.

In summary, the velocity distribution in the material removal area during one vibration cycle is shown in Figure 7. As shown in Figure 7a,c, when *θ* = 0° and *θ* = 180°, the velocities are distributed in the contact area and located in the middle of the material removal area, corresponding to Figure 6a,c, respectively. And at this time, there is *V_Z_* = *V_ZH_*, and the velocity *V* in the contact area is the combined velocity of *V_S_*, *V_W_*, and *V_Z_*. As shown in Figure 7b,d, when *θ* = 90° and *θ* = 270°, the velocities are distributed in the contact area and located on both sides of the material removal area, corresponding to Figure 6b,d, respectively. And at this time, there is *V_Z_* = 0, and the velocity *V* in the contact area is the combined velocity of *V_S_* and *V_W_*.

### 3.2. Pressure Distribution in Material Removal Area

In the 1880s, the German physicist Hertz proposed a theoretical formulation about the contact of elastomers, known as the Hertz contact theory [23]. According to Hertz, when pressure acts on two elastic objects, compressed, local elastic deformation occurs in the contact area. The elastic contact theory has the following three assumptions:

(1) There is only elastic deformation of two objects in contact and the deformation is following Hooke’s law [24].

During the TCG of optical components, the contact wheel is in elastic contact in the contact area. The local plastic deformation in the contact area is extremely small compared to the contact wheel diameter. The thickness of the abrasive belt substrate is also small, and the degree of its plastic deformation that occurs is relatively small. Therefore, it can be assumed that the contact between the contact wheel, the optical element, and the abrasive belt is following the elastic Hertz contact theory.

(2) The pressure acting on two objects is distributed perpendicular to the contact area.

The contact wheel is connected to the support shaft using bearings and is supplied with a stable contact pressure by a cylinder. During the actual machining process, the contact wheel is always in normal contact with the optical element to ensure that the pressure acting on the two objects is distributed perpendicular to the contact area.

(3) The deformation of the contact area is extremely small relative to the radius of the contact wheel.

The hardness and Young’s modulus of the contact wheel are small relative to the optical components, and the contact pressure provided by the cylinder during the actual machining process is small. Therefore, the deformation of the contact area is extremely small relative to the radius of the contact wheel.

In summary, the contact between the contact wheel and the optical components during the TCG process follows the Hertz contact theory.

The Hertz contact model between the contact wheel and the optical component is shown in Figure 8 [25]:

According to the Hertz contact theory, the pressure in the contact area can be expressed as:(8)$$P(x)=\left\{ \begin{array}{c} \frac{2F}{\pi a^{2}L_{W}}\sqrt{a^{2}-x^{2}},\left|x\right|\leq a\\ \;0\;,\left|x\right|>a \end{array}\right.$$
where *F* is the normal contact force between the contact wheel and the workpiece, *a* is half of the width of the contact area, *L_W_* is the width of the contact wheel and also the length of the contact area, and *x* is the distance from the current point to the contact center.

The width of the contact area can be expressed as:(9)$$a=\sqrt{\frac{4FR^{*}}{\pi E^{*}L_{W}}}$$
where *R*^∗^ is the equivalent radius and *E*^∗^ is the equivalent modulus of elasticity.
(10)$$R^{*}=\frac{1}{R_{1}}+\frac{1}{R_{2}}$$
(11)$$E^{*}=\frac{1}{\frac{1-v_{1}^{2}}{E_{1}}+\frac{1-v_{2}^{2}}{E_{2}}}$$
where *R*_1_ and *R*_2_ are the radii of the curvature of the contact wheel and the workpiece, respectively. *v*_1_ and *v*_2_ are the Poisson’s ratios of the contact wheel and the workpiece, respectively. *E*_1_ and *E*_2_ are the moduli of elasticity of the contact wheel and the workpiece, respectively.

In summary, the pressure distribution in the material removal area during one vibration cycle is shown in Figure 9. As shown in Figure 9a,c, when *θ* = 0° and *θ* = 180°, the pressure is distributed in the contact area and located in the middle of the material removal area, corresponding to Figure 6a,c, respectively. As shown in Figure 9b,d, when *θ* = 90° and *θ* = 270°, the pressure is distributed in the contact area and located on both sides of the material removal area, corresponding to Figure 6b,d, respectively. The pressure shows a distribution trend of high in the middle and low at both ends along the *X*-axis origin and the same distribution trend along the *Y*-axis direction.

### 3.3. Degradation of Removal Efficiency Due to Abrasion of Abrasive Belt

As the abrasive belt grinding process proceeds, the reaction force of grinding causes the belt to wear, resulting in the decay of the belt grinding capacity. The degradation of the grinding capacity of the abrasive belt will have a large impact on the efficiency of the TCG removal function [26]. The main forms of abrasive belt wear are abrasive grain shedding, clogging, and passivation [27,28,29]. Abrasive passivation causes significant changes in the geometry of the grains, including an increase in the negative front angle, a decrease in the height of the grain, and an increase in the top area of the grains [30,31]. The passivation process of abrasive belt grains during TCG is shown in Figure 10.

A section of the abrasive belt was sampled, and the state of the abrasive grains before, during, and after grinding of the belt was observed using a KEYENCE (Osaka, Japan) VHX-600 ultra-deep field microscope, as shown in Figure 11. As shown in Figure 11a, the left side of the abrasive belt shows the before-grinding state, the middle shows the during-grinding state, and the right side shows the after-grinding state. As shown in Figure 11b, the grinding grains are full and uniform before grinding, with a maximum height of 34.1 μm. As shown in Figure 11c, the grinding grains become less and less uniform during grinding, with a maximum height of 25.9 μm. As shown in Figure 11d, there are fewer grains after grinding and the uniformity of grains is poor, with a maximum height of 17.7 μm. Therefore, the sharp passivation wear of the abrasive belt in the grinding area during the TCG of optical components can have a large impact on the removal function efficiency.

T.O. Mulhearn et al. [32], based on abrasive belt grinding grain passivation experiments, concluded that the function describing abrasive belt wear is in the negative exponential form. They further derived that the total mass of material *M_n_* removed by the *n*th pass of the same abrasive belt through the grinding area can be expressed as:(12)$$M_{n}=e^{-\beta n}M_{\infty }$$
where *β* is the degradation factor, and *M_∞_* is the maximum material removal.

### 3.4. Removal Function Model of TCG

According to Equation (1), combined with Equations (6), (8), and (12), the removal function of TCG can be expressed as:(13)$$H(n)=e^{-\beta n}K\int_{-a}^{a}\int_{0}^{\Omega }\frac{2F}{\pi a^{2}L_{W}}\sqrt{(a^{2}-x^{2})\left[{\left(2\pi fr\cos(\theta )\right)}^{2}+{\left(V_{S}+V_{W}\right)}^{2}\right]}d\theta dx$$
where Ω is the rotation angle of point A.

The normalized theoretical removal function of TCG obtained from the simulation is shown in Figure 12. As shown in Figure 12b,c, the shape of the removal function is the axially (*Y*-axis direction) symmetric and circumferentially (*X*-axis direction) asymmetric prismatic cone. The horizontal section of the removal function shrinks in the axial and circumferential directions along the depth.

## 4. Experimental Setup

### 4.1. Experimental Equipments

#### 4.1.1. Experimental Equipment of TCG

Based on the principle of TCG, a TCG device was developed in our laboratory. The device is fixed on a 6-axis gantry machine, as shown in Figure 13, and the experiments in this paper were carried out on this TCG machine. The TCG device is mainly composed of three parts: a base, cylinder, and vibration mechanism. The entire device is fixed on the sliding plate of the 6-axis gantry machine tool through the base, controlled by the CNC system of the machine tool, and has the ability to process plane and aspheric optical components. The vibration mechanism and the base are connected by guide rails, and the cylinder controls the reciprocating motion of the vibration mechanism.

#### 4.1.2. Experimental Equipment of Measurement

After the removal function experiments, the removal functions were extracted using a self-developed high-precision, non-contact coordinate profiler [33]. Its main technical parameters are shown in Table 1, and the measurement process is shown in Figure 14. The non-contact coordinate profiler uses a spectral confocal sensor as a probe to obtain accurate information about the distance between a probe and a workpiece. The probe is better adapted to the surface roughness of a workpiece than the wavefront interferometer. Therefore, when using this profiler for removal function profile measurement, its uncertainty is low and meets the measurement accuracy requirements.

Before and after the shaping experiment, the surface shape error was measured using a Zeiss (Oberkochen, Germany) ACCURA II coordinate measuring machine (CMM). The surface roughness of the optical element was measured using the JITAI KEYI (Beijing, China) TR200 surface roughness meter (SRM).

### 4.2. Materials and Methods

In this study, the grinding specimen was ZERODUR^®^ glass-ceramic, an inorganic, non-porous lithium aluminum silicon oxide glass-ceramic produced by Schott AG (Mainz, Germany). The physical and mechanical properties of the material are shown in Table 2 [34]. The specimen size is 200 × 200 × 50 mm^3^.

The contact wheel used in this study had a diameter of 100 mm and a width of 20 mm. It consisted of an inner aluminum alloy wheel and an outer layer of rubber with hardness HA70. The abrasive belt model selected for this study was 461X, manufactured by 3M (Saint Paul, MN, USA). This is a thin-film polishing belt with SiC as the abrasive grains, with a thickness of 3MIL and an average abrasive grain size of 9 μm (about 1700 #) and 15 μm (about 900 #).

For the removal function experiments, the removal function was extracted using a fixed point and controlled residence time. To further demonstrate the robustness of the removal function model, two sets of experimental parameters were set. The residence times of the shaping experiment at different locations on the workpiece surface were obtained based on the extracted removal function and the initial surface shape of the glass-ceramic. The dwell time was solved for the machining area by cutting off the size of the removal function and the complementary edge size. In addition, considering the limited length of the open grinding abrasive belt, the single removal amount was set to 20% in the process software. The experimental parameters are shown in Table 3.

## 5. Results

### 5.1. Removal Function Experiment

The experimental results of the removal function under experimental value 1 are shown in Figure 15. From Figure 15, the contour measurements of the five groups of removal functions are macroscopically similar under the same experimental parameter conditions.

From Figure 16a–c, it can be seen that the 2D contour of the removal function is rectangular with a size of about 25 mm × 10 mm, which is consistent with the theoretical simulation results of the removal function in Figure 12.

As shown in Figure 16b, the experimental removal function and the theoretical removal function fit well for the circumferential contour of the removal function. As shown in Figure 16c, the experimental removal function and the theoretical removal function fit well for the axial contour of the removal function.

Figure 16d shows the 3D contour of the experimental removal function, and the overall contour is prismatic–conical with axial symmetry and circumferential asymmetry. The horizontal cross-section of the removal function shrinks in the axial and circumferential directions along the depth, and this conclusion is consistent with the normalized theoretical removal function in Figure 12.

The experimental results of the removal function under experimental value 2 are shown in Figure 17. As shown in Figure 17a, the experimental removal function and the theoretical removal function fit well for the circumferential contour of the removal function. As shown in Figure 17b, the experimental removal function and the theoretical removal function fit well for the axial contour of the removal function.

### 5.2. Shaping Experiment

The results of the shaping experiments for the glass-ceramic specimen are shown in Figure 18. Figure 18a shows the initial surface shape error of the specimen measured using the CMM of 6.497 μm PV and 1.318 μm RMS. Figure 18b shows the simulation processing results based on the theoretical removal function, and the surface shape error of the specimen converges to 5.079 μm PV and 1.051 μm RMS. Figure 18c shows the experimental processing results based on the actual removal function, and the surface shape error of the specimen converges to 5.397 μm PV and 1.115 μm RMS.

During the shaping process of the glass-ceramic specimen, TCG plays the role of polishing at the same time, and the surface roughness is improved. The surface roughness before and after TCG is measured via the nine-point sampling method. As shown in Figure 19a, the surface roughness before TCG is 271 nm Ra. As shown in Figure 19b, the surface roughness after TCG is 143 nm Ra. This result can also be visually obtained from Figure 20. In Figure 20b, the light transmission and reflection effect after TCG is significantly better than that before TCG in Figure 20a, which proves that the surface roughness is improved after the TCG process.

## 6. Discussion

In this study, a TCG removal function model was developed and the validity of the model was verified experimentally, which has potential application value in engineering.

As shown in Figure 16b, it can be seen that the experimentally obtained removal function circumferential contour is an asymmetric structure. This verifies the conclusion in Section 3 that the removal efficiency in the material removal area decreases sharply due to the abrasion of the abrasive belt [26], which leads to the asymmetric structure of the removal function circumferential contour.

As shown in Figure 16c, the axial contour of the removal function has an overall “trapezoidal” shape. The reason for the highest removal efficiency in the middle area is that this area is always in contact with the contact wheel during one vibration cycle. The two sides of the area are not always in contact with the contact wheel in one cycle because of the vibration of the contact wheel, and there is a speed change, as shown in Figure 6.

Figure 16 and Figure 17 show the experimental results for the two groups with different parameters. The theoretical removal function and the actual removal function of both sets of experiments are consistent. This conclusion proves the good robustness of the TCG removal function model established in this paper.

As shown in Figure 16b,c and Figure 17a,b, the theoretical removal function has some errors in both circumferential and axial contours, which may be caused by the existence of tilt between the contact wheel and workpiece contact, resulting in deviations between the actual material removal area pressure distribution and the theory.

From Figure 18, by comparing the simulated machining results with the experimental machining results, the difference between the two surface shape errors is 0.318 μm PV and 0.064 μm RMS. The results demonstrate that the simulated machining based on the theoretical removal function and the experimental machining based on the actual removal function are in good agreement, and the simulated machining can guide the actual machining, which also further proves the effectiveness of the removal function model in this paper. The reason for the difference between the simulated and actual machining results is mainly due to the extraction error of the removal function.

Improving the accuracy of the grinding stage is important for the manufacture of large-diameter optical components. In this paper, based on the idea of deterministic processing, a TCG removal function model was established, and the validity and robustness of the model were verified via experiments. The results of the study show the great potential of TCG in the processing of large-diameter optical components.

## 7. Conclusions

To improve the processing quality and determinism of abrasive belt grinding and to realize the application of abrasive belt grinding in the field of optical processing, this paper applied the TCG of the abrasive belt as a processing tool to the deterministic processing of optical components based on the idea of CCOS deterministic processing. This paper introduced the principle of TCG, established the removal function model of TCG, and carried out removal function experiments and shaping experiments. The main conclusions obtained are as follows:
(1)Based on the Preston equation, the TCG removal function model is established. The velocity distribution and pressure distribution in the contact area will show periodic changes with the vibration of the contact wheel, and the material removal efficiency is highest in the area that is always in contact with the contact wheel in one cycle. The abrasive belt wear has a large impact on the contour of the removal function, resulting in an asymmetric structure of the removal function in the circumferential direction. The 3D contour of the removal function shows an axially symmetric and circumferentially asymmetric prismatic cone. The horizontal section of the removal function shrinks in the axial and circumferential directions along the depth.(2)Based on the removal function experiments, the validity and robustness of the TCG removal function model is verified. Through the comparative analysis of the experimental removal function and the theoretical removal function, the axial and circumferential contours of the theoretical removal function and the experimental removal function are fitted well. It is worth noting that there are some deviations between the experimental removal function and the theoretical removal function, which may be caused by the tilt of the contact wheel and the workpiece contact, resulting in the deviation in the pressure distribution in the actual material removal area and the theory.(3)Based on the TCG shaping experiment, it is verified that the simulation of the theoretical removal function can guide the actual processing. Based on the 20% single material removal, the theoretical and actual surface shape errors converge from 6.497 μm PV and 1.318 μm RMS to 5.079 μm PV and 1.051 μm RMS and 5.397 μm PV and 1.115 μm RMS, respectively. By comparing the simulated machining results with the experimental machining results, the difference between the two surface shape errors is 0.318 μm PV and 0.064 μm RMS. The reason for the difference between the simulated and actual machining results is mainly due to the extraction error of the removal function. The results verify that the theoretical removal function model established in this paper can guide the actual TCG shaping experiments.(4)TCG takes into account polishing characteristics while shaping. During the TCG shaping experiments, the surface roughness of glass-ceramic is improved from 271 nm Ra to 143 nm Ra, proving that the surface roughness of TCG is improved while shaping.


## Figures and Tables

**Figure 1 micromachines-14-01384-f001:**
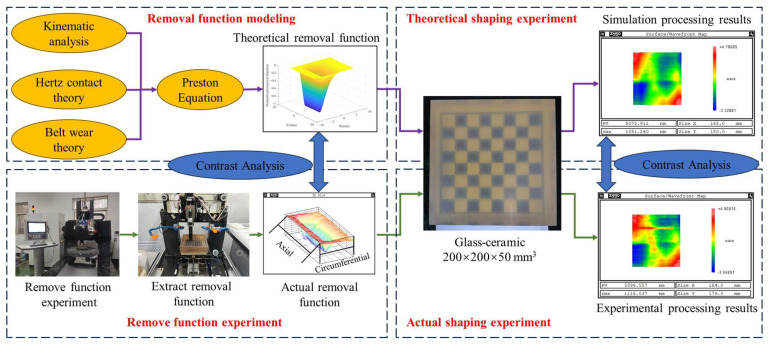
The research idea of this paper.

**Figure 2 micromachines-14-01384-f002:**
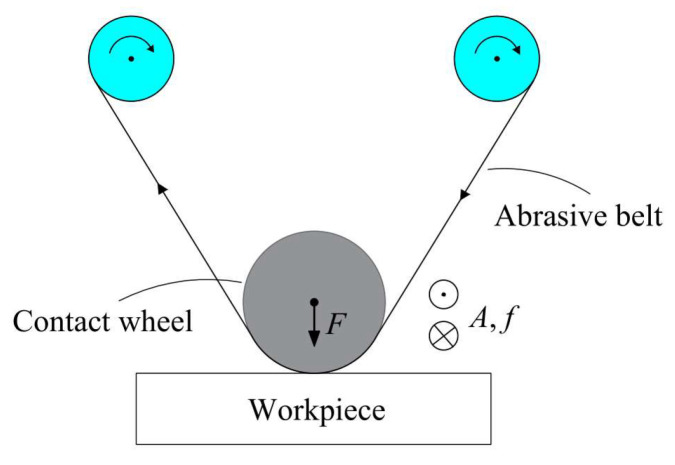
Principle of TCG.

**Figure 3 micromachines-14-01384-f003:**
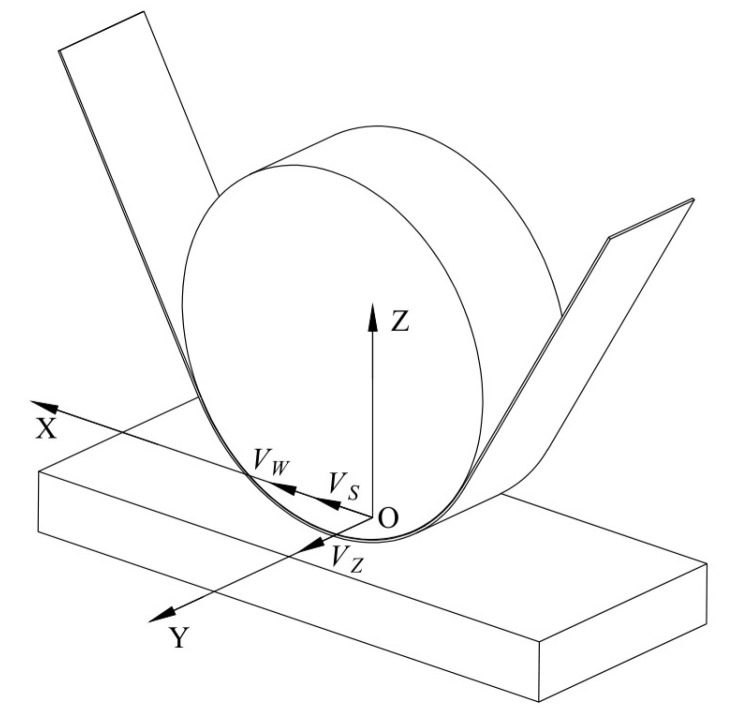
Velocity distribution in material removal area.

**Figure 4 micromachines-14-01384-f004:**
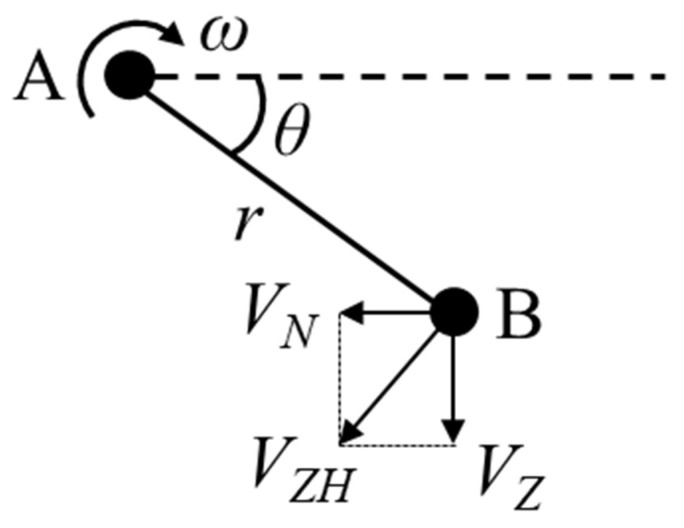
Principle diagram of eccentric vibration mechanism.

**Figure 5 micromachines-14-01384-f005:**
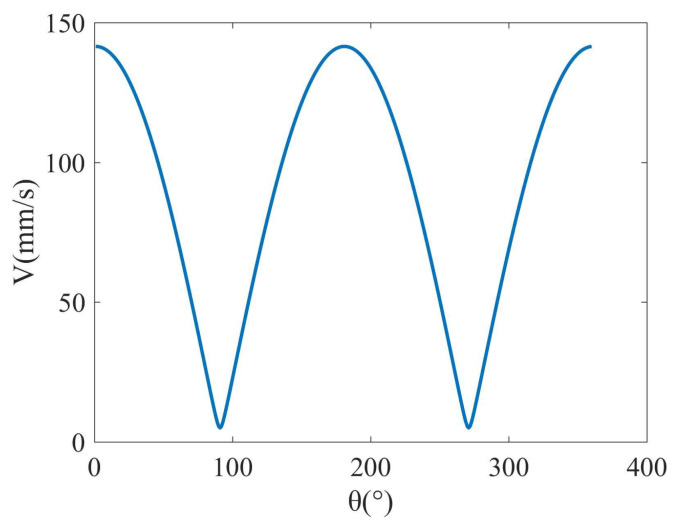
Speed variation in abrasive belt grains in one vibration cycle.

**Figure 6 micromachines-14-01384-f006:**
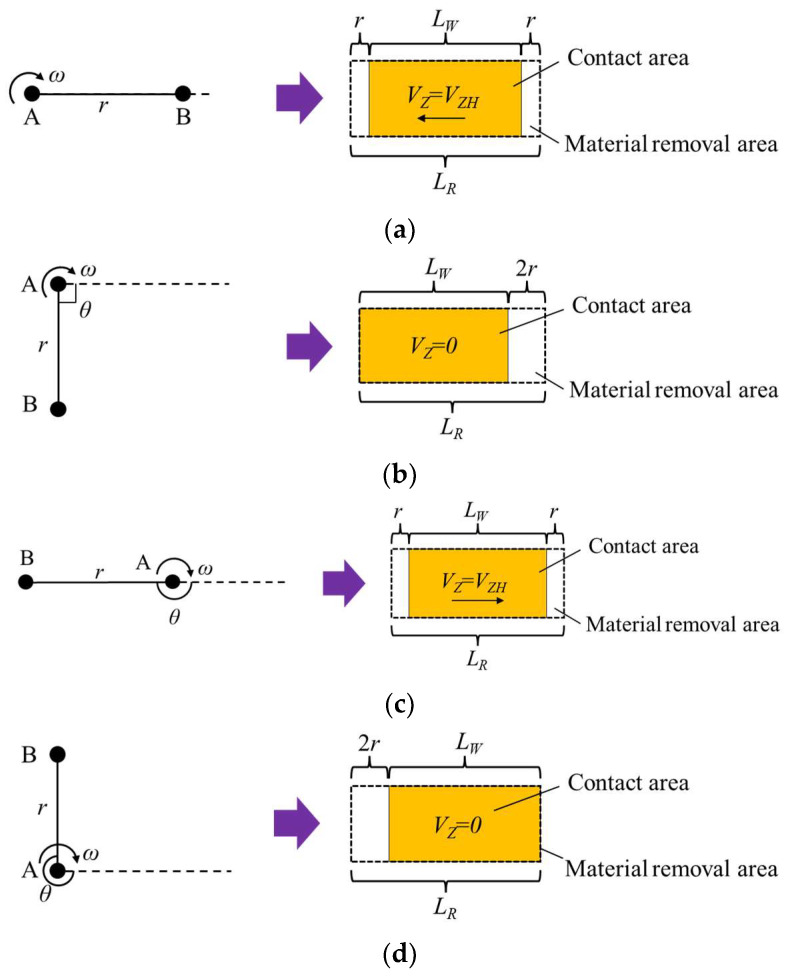
The rotation angle of the eccentric mechanism during one vibration cycle corresponds to the position of the contact area in the material removal area: (**a**) *θ* = 0°; (**b**) *θ* = 90°; (**c**) *θ* = 180°; (**d**) *θ* = 270°.

**Figure 7 micromachines-14-01384-f007:**
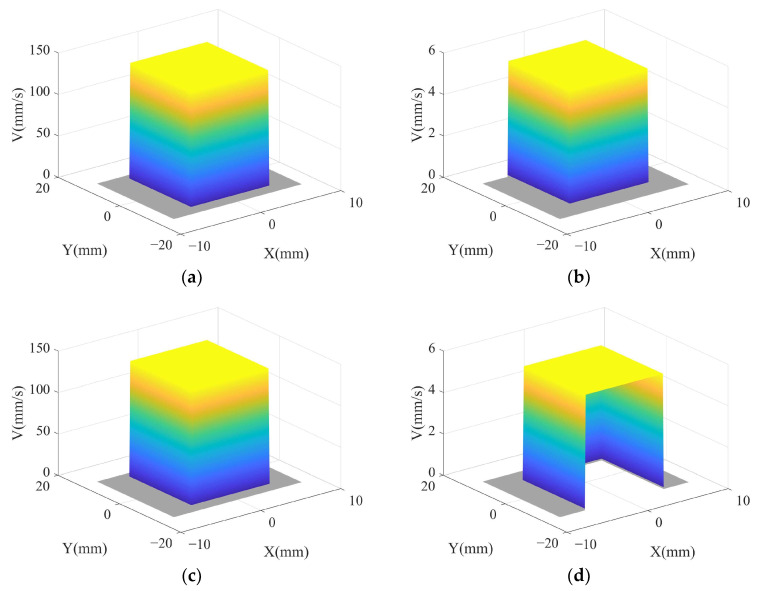
Velocity distribution of material removal area in one vibration cycle: (**a**) *θ* = 0°; (**b**) *θ* = 90°; (**c**) *θ* = 180°; (**d**) *θ* = 270°.

**Figure 8 micromachines-14-01384-f008:**
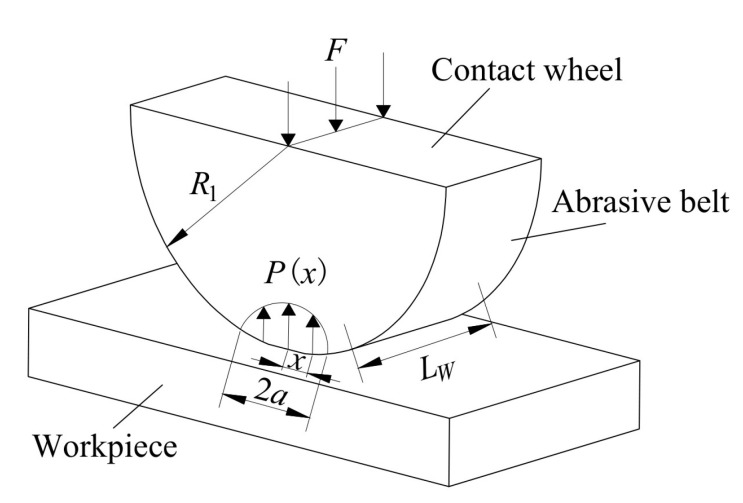
Hertz contact model of TCG.

**Figure 9 micromachines-14-01384-f009:**
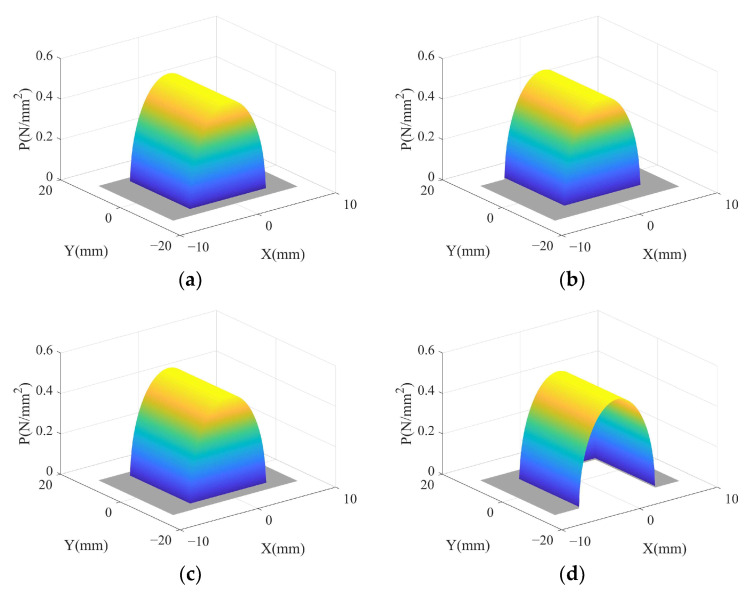
Pressure distribution of material removal area in one vibration cycle: (**a**) *θ* = 0°; (**b**) *θ* = 90°; (**c**) *θ* = 180°; (**d**) *θ* = 270°.

**Figure 10 micromachines-14-01384-f010:**
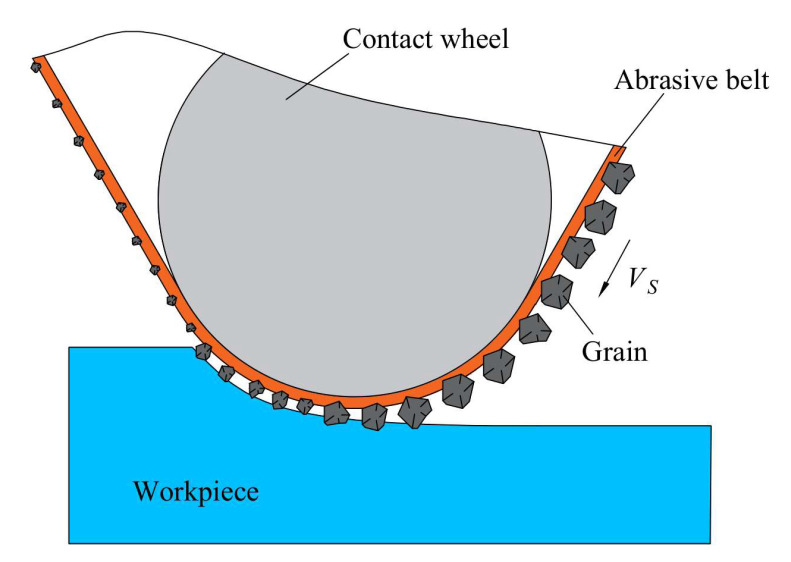
Passivation process of abrasive belt grains.

**Figure 11 micromachines-14-01384-f011:**
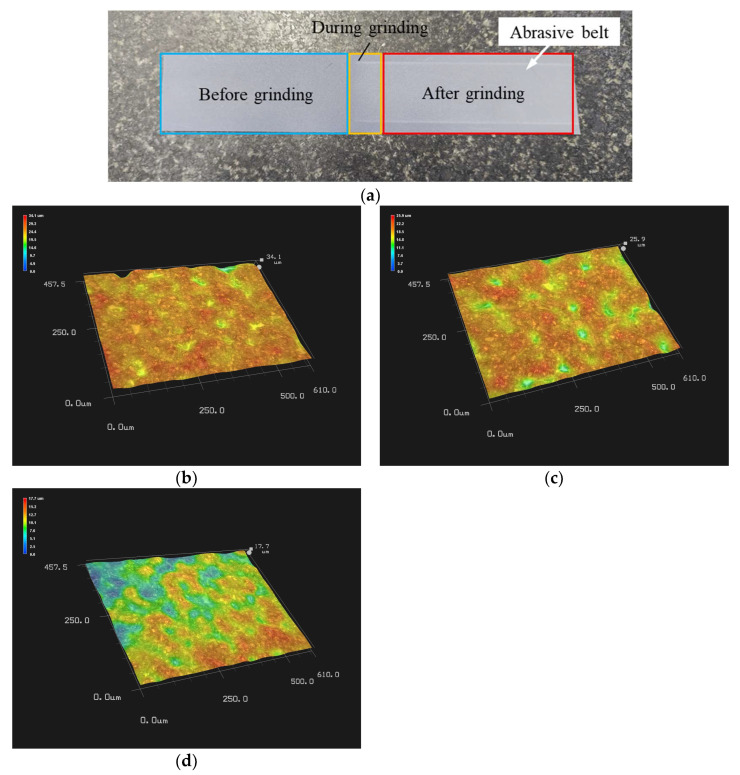
The state of the grains: (**a**) abrasive belt; (**b**) before grinding; (**c**) during grinding; (**d**) after grinding.

**Figure 12 micromachines-14-01384-f012:**
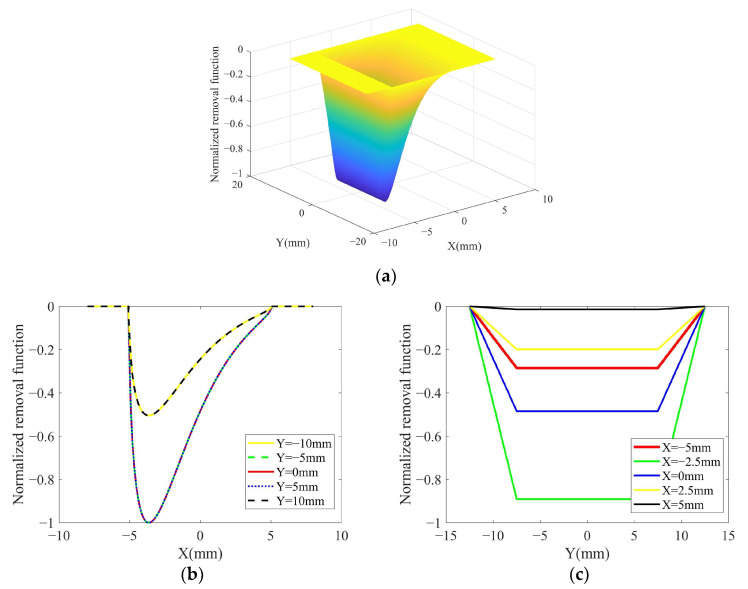
Normalized theoretical removal function of TCG: (**a**) 3D contour; (**b**) 2D contour in *X*-axis direction; (**c**) 2D contour in *Y*-axis direction.

**Figure 13 micromachines-14-01384-f013:**
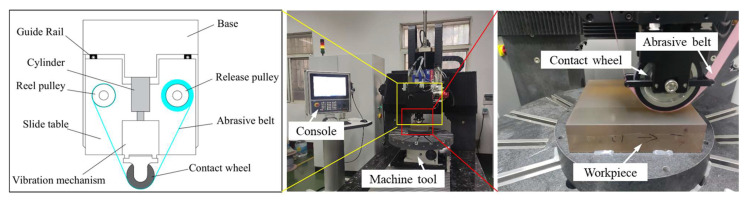
TCG machine.

**Figure 14 micromachines-14-01384-f014:**
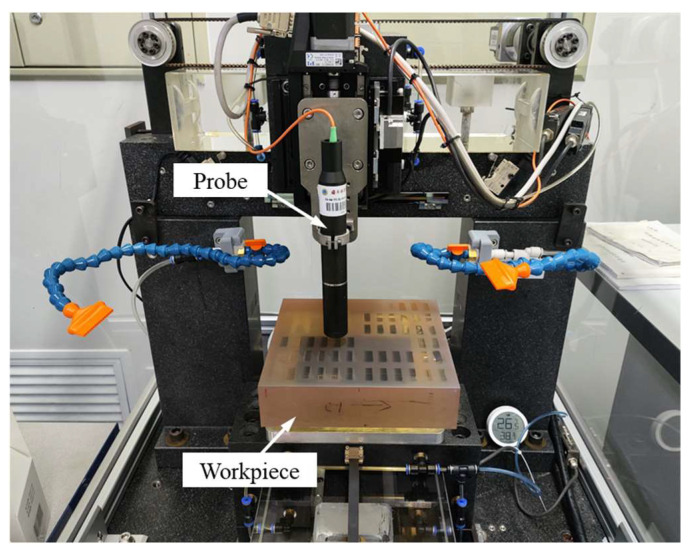
Non-contact coordinate profiler.

**Figure 15 micromachines-14-01384-f015:**
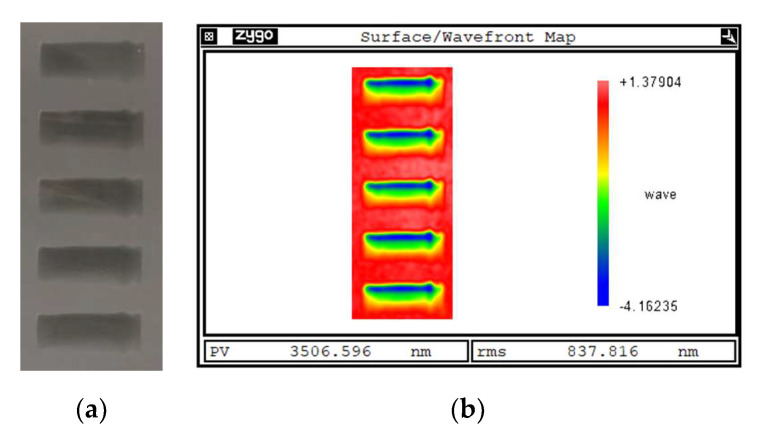
Experimental results of removal functions under experimental value 1: (**a**) actual removal function; (**b**) contour measurement results.

**Figure 16 micromachines-14-01384-f016:**
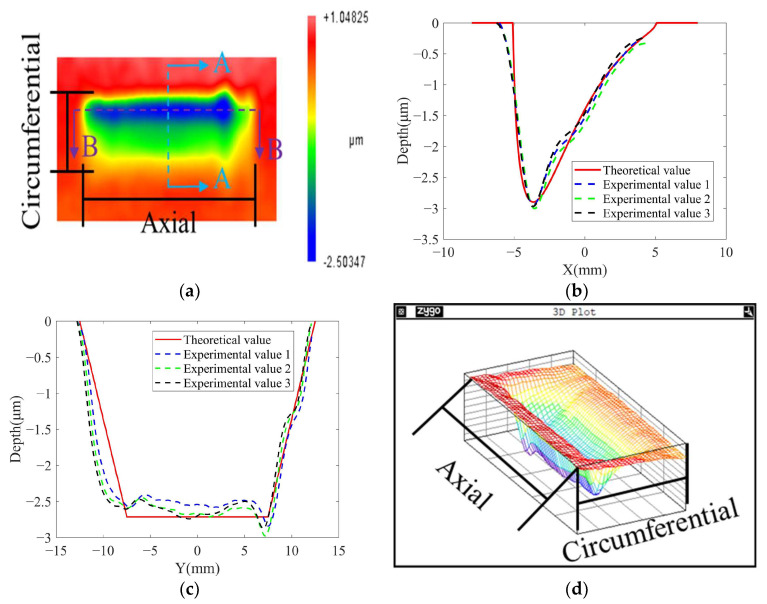
Comparison between theoretical removal function and actual removal function under experimental value 1: (**a**) 2D contour of experimental removal function (A-A is the circumferential section view; B-B is the axial section view); (**b**) circumferential contours; (**c**) axial contours; (**d**) 3D contour of experimental removal function.

**Figure 17 micromachines-14-01384-f017:**
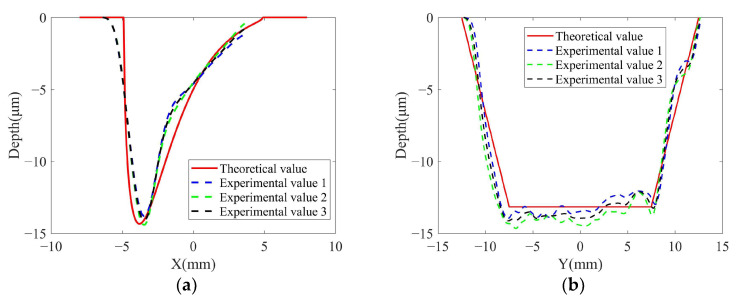
Comparison between theoretical removal function and actual removal function under experimental value 2: (**a**) circumferential contours; (**b**) axial contours.

**Figure 18 micromachines-14-01384-f018:**
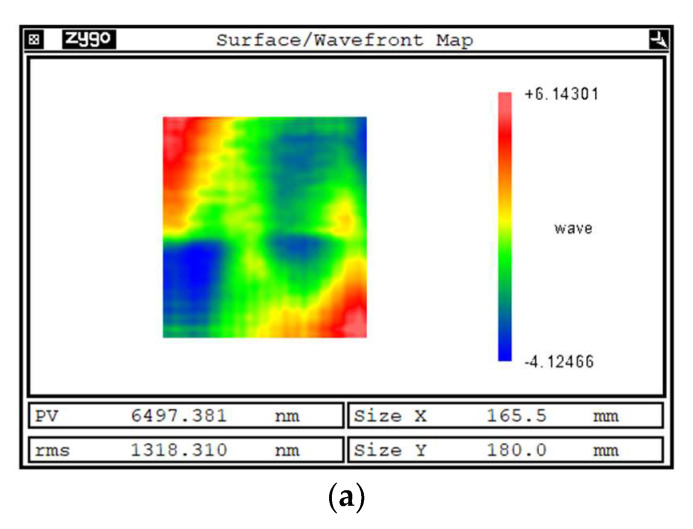
Shaping results of glass-ceramic specimen: (**a**) initial surface shape error; (**b**) simulation processing results based on theoretical removal function; (**c**) experimental processing results based on the actual removal function.

**Figure 19 micromachines-14-01384-f019:**
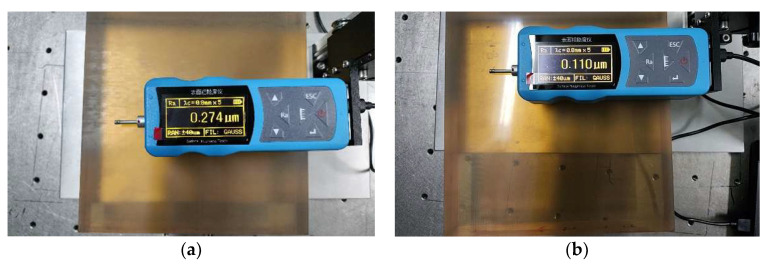
Measurement results of surface roughness: (**a**) before TCG; (**b**) after TCG.

**Figure 20 micromachines-14-01384-f020:**
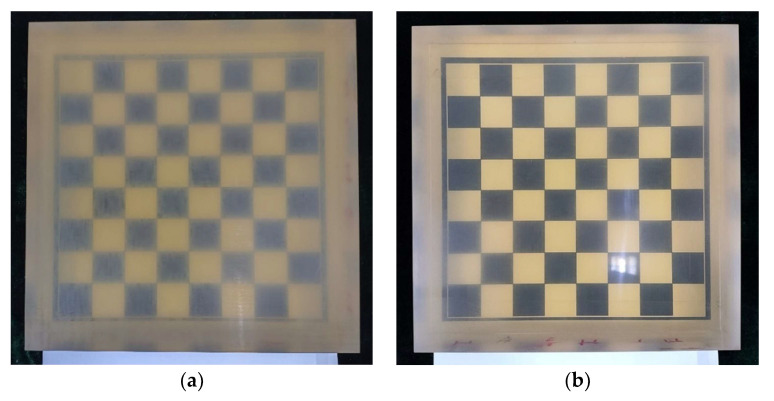
Surface roughness comparison: (**a**) before TCG; (**b**) after TCG.

**Table 1 micromachines-14-01384-t001:** Main technical parameters of non-contact coordinate profiler.

Properties	Value
Measuring range (mm)	200 × 200 × 52
Spatial resolution X/Y direction (μm)	1.7
Spatial resolution Z direction (nm)	22
Maximum local deviation angle (°)	±14°
Spatial uncertainty (nm)	<181.91 PV
Low steepness uncertainty (nm)	<38.46 PV

**Table 2 micromachines-14-01384-t002:** The physical and mechanical properties of ZERODUR^®^.

Properties	Value
Young’s modulus *E* (GPa)	90.3
Poisson’s ratio *μ*	0.24
Density *ρ* (g/cm^3^)	2.53
Knoop Hardness *HK* 0.1/20	620
Thermal expansion coefficient 1/*K*	10^−8^

**Table 3 micromachines-14-01384-t003:** Experimental parameters.

Experimental Parameters	Remove Function Experimental Value 1	Remove Function Experimental Value 2	Shaping Experimental Value
Grain Size (μm)	9	15	9
Updating Speed (mm/s)	5	4	5
Vibration Frequency (Hz)	9	8	9
Contact Pressure (MPa)	0.25	0.2	0.25
Residence Time (s)	90	60	—

## Data Availability

The data presented in this study are available upon request from the corresponding author. The data are not publicly available because they are part of an ongoing study.

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
