# Peer review of "Modeling and Experimental Verification of Time-Controlled Grinding Removal Function for Optical Components"

_micromachines, 2023, doi:10.3390/mi14071384_

Round 1
Reviewer 1 Report
This paper studies the time-controlled grinding removal function for optical components, which has potential application value in engineering. In order to meet the requirements of high-quality publication of the journal (micromachines), it is recommended to consider the following suggestions.
1) The background section of the abstract needs to be simplified..
2) "The results show that the surface shape error converges from 6.497 μm PV, 1.318 μm RMS to 5.397 μm PV, 1.115 μm RMS" ? Please Check!
3) It is best to have a schematic diagram for the first section to facilitate readers' understanding.
4) What is the basis for selecting parameter values in Table 3?
5) The mehtod proposed in this paper needs to be compared with the previous literature, otherwise it cannot reflect innovation.
6) The discussion needs to be divided into a separate section.
7) Line 386 "5. Results and Discussion" ,Please Check.
Reviewer 2 Report
1. Fig. 6 and 8 are same. Fig. 8 is supposed to describe pressure distribution.
2. The results do not depend on the Y coordinate in the Figures 6, 8, 11. They can be shown in 2D plots.
I do not have further comments. The theoretical and experimental results are consistent. It seems to be useful research for the particular application.
nil
Reviewer 3 Report
This study develops the idea of deterministic machining of materials by time-controlled grinding. The theoretical removal model was proposed which are in good coincidence with the experiments on shaping of large-diameter optical glass-ceramic components. The following issues should be clarified:
1) Is the developed model applicable to other optical materials? Also I recommend the variation of remove function experimental parameters to check the robustness of the proposed model.
2) It is necessary to indicate the software used for theoretical calculations as well as the model of the ultra-deep field microscope (line 241).
3) The main three parts (base, cylinder, vibration mechanism) of TCG device should be clearly indicated in Fig. 12.
4) What is PV unit in Table 1?
5) The source of ZERODUR® glass-ceramics properties (Table 2) should be referenced.
Round 2
Reviewer 1 Report
The authors have addressed all my concerns.
Reviewer 3 Report
The authors adequately answered all the questions raised and the quality of the manuscript was improved.